# Enhanced Electrical Injury Using Triangular Interdigitated Electrodes for Catheter-Based Irreversible Electroporation

Dong-Jin Lee [1,2] and Dae Yu Kim [1,2,3,*]

1    Inha Research Institute for Aerospace Medicine, Inha University, Incheon 22212, Republic of Korea; djlee@inha.ac.kr
2    Center for Sensor Systems, Inha University, Incheon 22212, Republic of Korea
3    Department of Electrical and Computer Engineering, College of Engineering, Inha University, Incheon 22212, Republic of Korea
*    Correspondence: dyukim@inha.ac.kr

**Featured Application: The triangular interdigitated electrodes proposed in this paper could potentially provide an alternative strategy for designing interdigitated electrodes for catheter-based irreversible electroporation.**

**Abstract:** Irreversible electroporation (IRE) is a promising nonthermal ablation technique that uses high-voltage electrical pulses to create permanent pores in the cell membrane of target tissue. Recently, endoscopic IRE with catheter-based electrodes has attracted significant attention as a potential alternative tool for gastrointestinal tumors, but it has been challenged owing to the limited electric field distribution in an in-plane electrode configuration, in which rectangular interdigitated electrodes (IDEs) are commonly used. Herein, we report an enhanced electrical injury in tissue using triangular IDEs that cause strong electric fields to be induced at the tip of the electrode fingers. A set of 10 pulses with a duration of 100 µs and a frequency of 1 Hz were delivered to the tissue, and a finite element method was used to calculate the electrical injury in the gastrointestinal model. The probability of cell death by electrical injury at the triangular IDEs increases by approximately 10 times compared to that of conventional rectangular IDEs at the same electrode distance. These results could potentially pave the way toward designing electrodes in catheter-based IRE devices.

**Keywords:** irreversible electroporation; triangular interdigitated electrodes; electrical injury

## 1. Introduction

Irreversible electroporation (IRE) is a nonthermal ablation technique that uses high-voltage electrical pulses to create permanent pores in the cell membrane of target tissue, leading to cell death [1,2]. Currently, IRE is considered one of the most efficient methods of destroying tumors in various organs, including the liver, pancreas, kidney, lung, brain, and prostate [3–12]. In clinical use, needle-shaped electrodes puncture the target tissue surrounding the tumor to deliver electrical pulses with the electric field strength of a few kV/cm [13]. However, this electrode configuration is unsuitable for vulnerable gastrointestinal tracts with several hollow organs [14].

Catheter-based IRE presents a promising alternative tool for effective treatment of gastrointestinal tumors or undesired tissues [14,15]. In the clinical process, a catheter is inserted into the body and guided to the tumor site, where it delivers electrical pulses to the tumor. This method is suitable for preventing perforation or injury. Jeon et al. designed two types of catheters, the needle-type and the basket-type, and investigated the feasibility and effectiveness of endoscopic IRE therapy on hollow viscous tissues [16]. Further, Rohan et al. demonstrated the feasibility of IRE based on a three-electrode tubular catheter in an experimental liver model in both ex vivo and in vivo studies [17]. In addition, interdigitated electrodes (IDEs) have been studied for their ability to reduce the required voltage and localize the electric fields at the

desired region [13,14]. IDEs further allow the electrode distance to be closer and thinner due to simple in-plane geometry [18,19]. Most of the finger structures of IDEs reported so far have a rectangular cross-section, and most studies in this area have focused on the effects of the finger's width, length, and period have been conducted [20,21]. Similarly, studies on IDEs used for electroporation have mainly concentrated on electrodes with a rectangular cross-section and various pulse settings (duration, total pulse number, frequency, voltage), but there has been no study on finger shape according to the authors' knowledge [13,22–24]. Therefore, further improvement in IDEs to enhance the electric field strength is required to increase the practicability of catheter-based IRE.

In this work, we propose triangular IDEs for catheter-based IRE by demonstrating electrical injury in a gastrointestinal model. The triangular IDEs allow strong electric fields to be induced at the tip of the electrode fingers and are simply fabricated using laser-induced graphene technology on a flexible polyimide film [18]. A three-dimensional finite element model based on COMSOL Multiphysics 5.4 software was adopted to investigate the enhancement in the electric field through the electrode design. With triangular IDEs, the probability of cell death due to electrical injury (EI) in tissue increased around 10 times compared to that of conventional rectangular IDEs at the same electrode distance. These results could indicate an alternative strategy for designing IDEs for catheter-based IRE.

## 2. Materials and Methods

### 2.1. Model of Gastrointestinal Tract

The gastrointestinal tract model, which included the mucosal layer (ML), submucosal layer (SL), and muscularis propria (MP), had a length of 40 mm and a thickness of 4 mm, as shown in Figure 1. The thicknesses of the ML, SL, and MP were 1, 1, and 2 mm, respectively. Each layer was considered to have isotropic and homogeneous electrical and thermal properties as reported in a previous study [14,25–27].

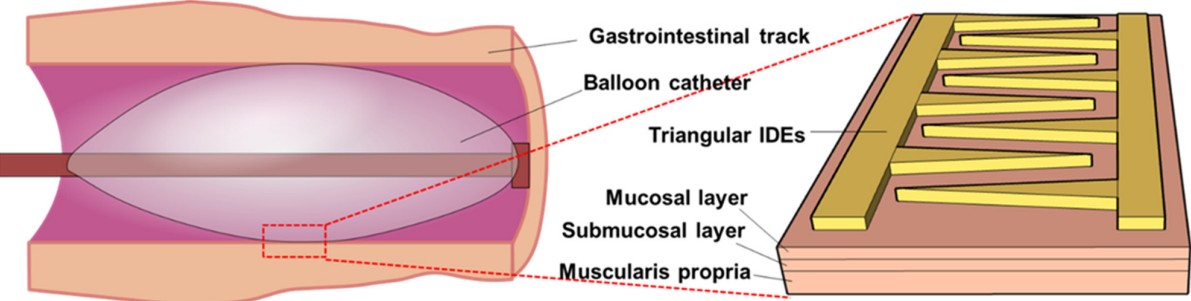

**Figure 1.** Model of gastrointestinal tract and triangular interdigitated electrodes (IDEs). The gastrointestinal track contains the mucosal layer, submucosal layer, and muscularis propria. Triangular IDEs are located on the outermost part of the balloon catheter and are in contact with the gastrointestinal track.

### 2.2. Electrode Configuration

The basic concept of the electrode design is shown in Figure 1. To compare the enhancement in the electric field, we designed two IDE configurations: (1) a conventional rectangular pattern and (2) a triangular pattern. The anode and cathode, which were both located inside the gastrointestinal track, both had a thickness of 0.2 mm.

### 2.3. Numerical Analysis

A three-dimensional finite element model of IRE was developed using COMSOL Multiphysics 5.4 software (COMSOL Inc., Palo Alto, CA, USA), which was used to investigate the electric field distribution created by the voltage applied to the electrodes and the

resulting electromagnetic heating from current flow. The electric field was determined by solving the governing Laplace equation [14]:

$$\nabla \cdot (\sigma \nabla \varphi) = 0, \tag{1}$$

where $\varphi$ is the voltage (V) and $\sigma$ is the electrical conductivity of the tissue (S/m).

The temperature distribution was solved using the modified Pennes' bioheat transfer equation [14,28]:

$$\nabla \cdot (k \nabla T) + \omega_b c_b (T_a - T) + Q_m + \sigma |E|^2 = \rho c_t \frac{\partial T}{\partial t}, \tag{2}$$

where $k$ is the thermal conductivity of the tissue (W/m $\cdot$ K), $T$ is the temperature (K), $\omega_b$ is the blood perfusion rate (1/s), $c_b$ is the specific heat capacity of blood (J/kg $\cdot$ K), $T_a$ is the arterial temperature (K), $Q_m$ is the metabolic heat generation in the tissue (W/m$^3$), $\rho$ is the tissue density (kg/m$^3$), and $c_t$ is the specific heat capacity of the tissue (J/kg $\cdot$ K). The joule heating term ($\sigma |E|^2$) was averaged over the entire duration of the treatment instead of calculating the heating from each individual pulse.

The tissue electrical conductivity was determined using the following equation to incorporate changes occurring due to electroporation and the thermal effect [14]:

$$\sigma_d(E, T) = \sigma_0 \left[1 + A flc2hs \left(normE - E_{delta}, E_{range}\right) + \alpha (T - T_0)\right], \tag{3}$$

where $\sigma_0$ is the baseline electrical conductivity, $A$ is the multiplication factor, $\alpha$ is the conductivity–temperature coefficient, $T_0$ is the physiological temperature, and $E_{delta}$ is the threshold electric field in electroporation. $flc2hs$ is a smoothed Heaviside function defined in COMSOL software, which changes from zero to one over the range of $\pm E_{range}$. $A$, $E_{range}$, and $E_{delta}$ are given in Table 1. The conductivity–temperature coefficient was taken as 0.02 (1/K) in this study. The value of 0.02 (1/K) was chosen from the literature [29,30].

**Table 1.** Electrical and thermal properties of the gastrointestinal tract and electrode.

| Material | Muscularis Propria | Submucosal Layer | Mucosal Layer | Electrode |
|---|---|---|---|---|
| Electrical conductivity (S/m) | 0.202 | 0.251 | 0.511 | $4.5 \times 10^7$ |
| $A$ | 1.5 | 1.8 | 1.8 | - |
| $E_{delta}$ (V/cm) | 500 | 600 | 600 | - |
| $E_{range}$ (V/cm) | 300 | 150 | 150 | - |
| Density (kg/m$^3$) | 1090 | 1027 | 1088 | 19,300 |
| Specific heat capacity (J/kg $\cdot$ K) | 3421 | 2372 | 3690 | 129 |
| Thermal conductivity (W/m $\cdot$ K) | 0.49 | 0.39 | 0.53 | 317 |
| References | [14,31–34] | [14,31–34] | [14,31–34] | - |

### 2.4. Probability of Cell Death Due to EI

The probability of cell death by EI during IRE was calculated using the Peleg–Fermi model, as follows [14]:

$$S = \frac{1}{1 + e^{\frac{E - E_c(n)}{A(n)}}} \tag{4}$$

where $S$ is the proportion of surviving cells after IRE and $E$ is the electric field strength. $E_c(n)$ and $A(n)$ are two functions that depend on the number of pulses:

$$E_c(n) = E_0 \exp(-k_1 n) \tag{5}$$

$$A(n) = A_0 \exp(-k_2 n) \tag{6}$$

where $E_0$, $A_0$, $k_1$, and $k_2$ are the regression coefficients that are related to the pulse duration and cell type, having values of 3996 V/cm, 1441 V/cm, 0.03, and 0.06, respectively, as described in the literature [35]. The probability of cell death due to EI can be calculated as follows:

$$Probability\ of\ EI\ (PEI) = 1 - S. \qquad (7)$$

Thus, a certain region where the PEI is greater than 0.90 is determined to be treated with IRE.

### 3. Results

The simulation models of the gastrointestinal tract and triangular IDEs are illustrated in Figure 1. The gastrointestinal tract was composed of the mucosal layer (ML), submucosal layer (SL), and muscularis propria (MP). The triangular IDEs were located inside the gastrointestinal tract. The electrical and thermal properties of gastrointestinal tract were based on the literature, and the electrode was set to be gold in the COMSOL built-in library (Table 1).

The electrical conductivity response according to the change in the temperature and electric field of the tissue is shown in Figure 2. The electrical conductivity was calculated through Equation (3) within the temperature range of 37–57 °C and the electric field range of 100–800 V/cm. The ranges of electrical conductivity change for the ML, SL, and MP used in this study were 0.511–1.638, 0.251–0.803, and 0.202–0.586 S/m, respectively. The above values were used in FEM simulation for the electric field distribution in the tissue and for probability of cell death due to EI (PEI). In addition, it should be noted that the influence of temperature on conductivity was small enough to be negligible compared with the change in electrical conductivity for electric field strength [33].

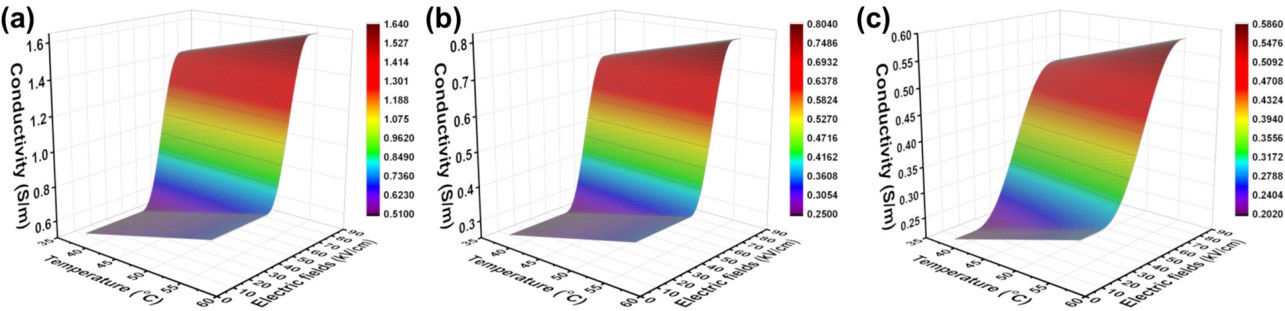

**Figure 2.** Electrical conductivity response of the (**a**) mucosal layer, (**b**) submucosal layer, and (**c**) muscularis propria during irreversible electroporation (IRE).

In order to investigate the effect of triangular IDEs, the electrical performances of rectangular and triangular IDEs with an identical electrode distance were compared. The rectangular IDEs had a finger length of 15 mm, finger width of 5 mm, electrode distance of 5 mm, and three fingers, as shown in Figure 3a. Two triangular IDEs were designed with (1) three fingers with a finger length of 15 mm, a base length of 5 mm, and a ~2 times longer electrode distance (d ≈ 11 mm) than rectangular IDEs as shown in Figure 3b; and (2) five fingers with a finger length of 15 mm, a base length of 5 mm, and an identical electrode distance (d ≈ 5 mm) of rectangular IDES as shown in Figure 3c. The voltage distribution for each electrode configuration is shown in Figure 3. A set of 10 pulses with a duration of 100 µs and a frequency of 1 Hz were delivered to the tissue, and the voltage of each pulse was set to be 1 kV. As shown in the voltage profile of the rectangular (d = 5 mm) and the triangular IDEs (d ≈ 11 mm, d ≈ 5 mm) at x = 20 mm and z = 3.5 mm in Figure 3d, the slopes of the rectangular IDEs (d = 5 mm) and the triangular IDEs (d ≈ 5 mm) in the region shaded in green were similar at 159.8 and 145.7, respectively, but the slope of the triangular IDEs (d ≈ 11 mm) was 83.2, showing a large difference.

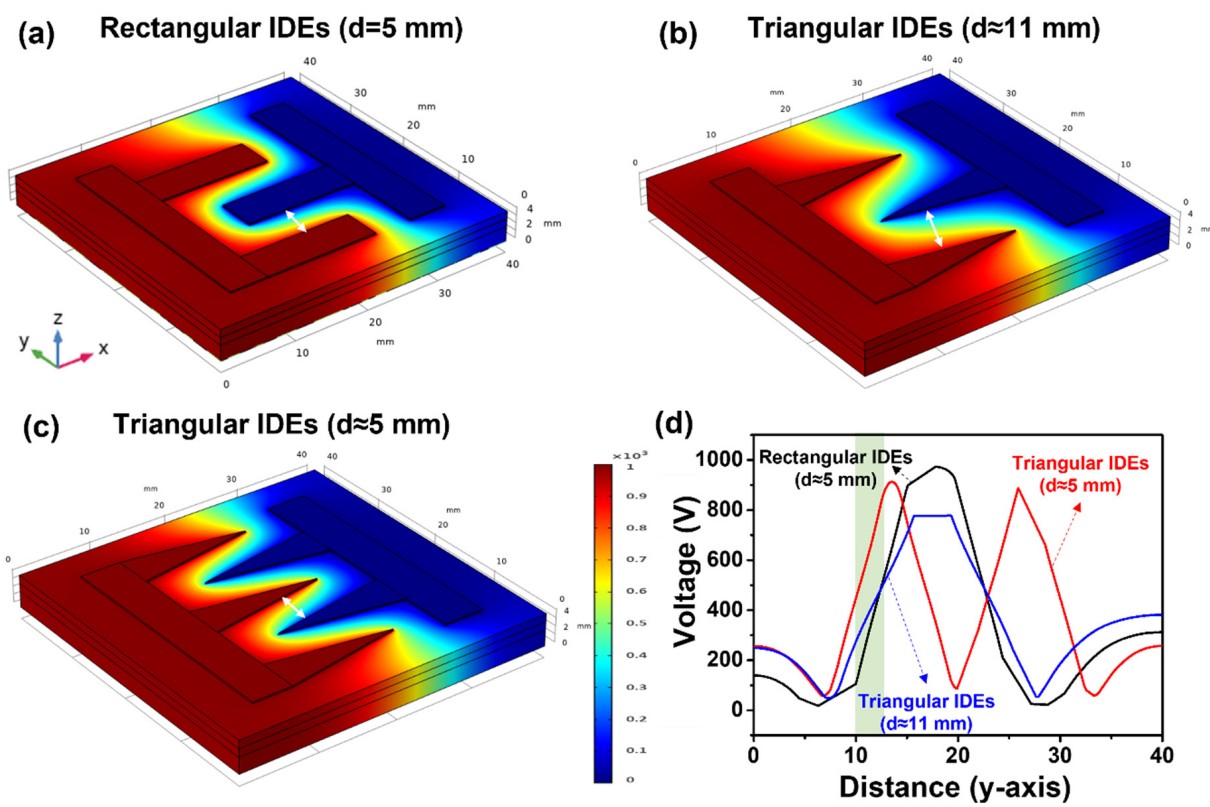

**Figure 3.** Voltage distribution of (**a**) rectangular IDEs with three fingers and electrode distance of 5 mm (d = 5 mm), (**b**) triangular IDEs with three fingers and electrode distance of 11 mm (d ≈ 11 mm), and (**c**) triangular IDEs with five fingers and electrode distance of 5 mm (d ≈ 5 mm) during IRE. The white arrow denotes the distance between electrodes. (**d**) The voltage variation in the rectangular (d = 5 mm) and the triangular IDEs (d ≈ 11 mm, d ≈ 5 mm) on y-axis at x = 20 mm and z = 3.5 mm.

The electric field distribution in the rectangular and triangular IDEs was evaluated as shown in Figure 4. The top panel shows the electric field distribution in the x-y plane view and the bottom panel shows the y–z cross-sectional view. The electric field was strongly focused on the edge of the rectangular IDE (d = 5 mm), and the maximum electric field was about $1.09 \times 10^6$ V/m. For the triangular IDE, a strong electric field was observed at the tip region, and the maximum magnitudes were about $2.54 \times 10^6$ V/m at d ≈ 11 mm and $3.17 \times 10^6$ V/m at d ≈ 5 mm. Compared to the triangular IDEs with a d ≈ 11 mm, the triangular IDEs with a d ≈ 5 mm had a smaller effective electrode spacing, so the magnitude of the electric field appears larger.

Figure 5 shows the distribution of PEI calculated from Equation (7) for the rectangular IDE (d = 5 mm), triangular IDE with three fingers (d ≈ 11 mm), and triangular IDE with five fingers (d ≈ 5 mm). From these results, a relative area ratio of PEI > 0.90 was calculated. The relative area ratio of PEI was defined as the ratio of the valid area of PEI > 0.90 to the total cross-sectional area. Figure 6 shows a comparison of the relative area ratio of PEI in rectangular IDEs (d = 5 mm), triangular IDEs with three fingers (d ≈ 11 mm), and triangular IDEs with five fingers (d ≈ 5 mm). The relative area ratios of the rectangular IDE (d = 5 mm), triangular IDE with three fingers (d ≈ 11 mm), and triangular IDE with five fingers (d ≈ 5 mm) were 0.18%, 1.45%, and 1.97%, respectively. The relative area ratio of the triangular IDEs with five fingers increased around 10 times compared to that of the conventional rectangular IDEs at the same electrode distance.

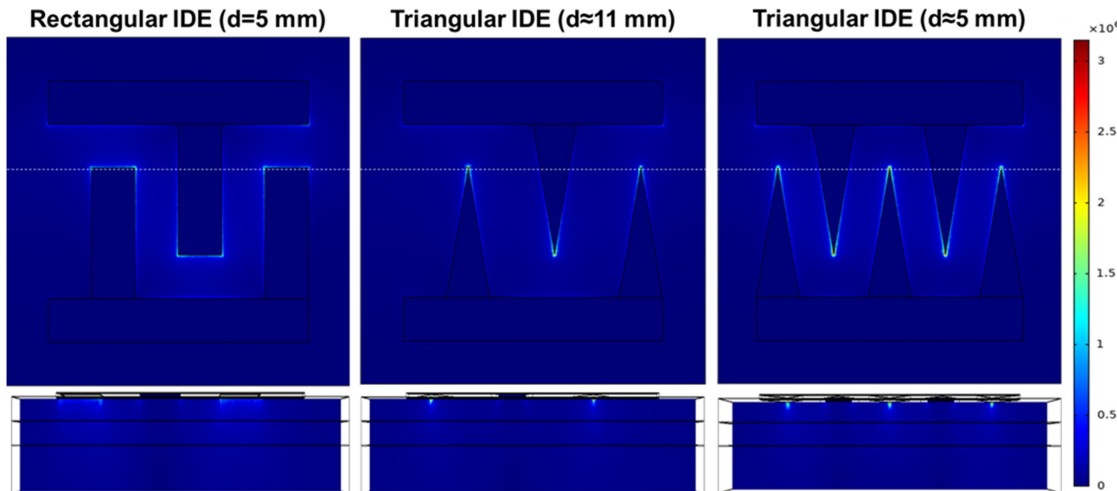

**Figure 4.** Effects of electrode configuration and distance on the distribution of electric fields in the x–y plane view (**top**) and y–z cross-sectional view (**bottom**). The white dashed line in the top view indicates the cutting region for the cross-sectional view.

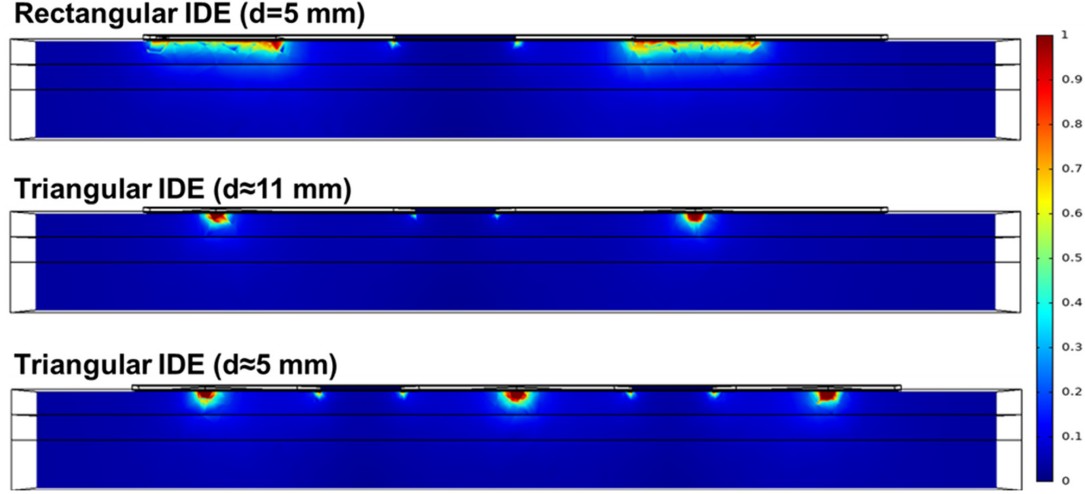

**Figure 5.** Probability of electrical injury of (**a**) rectangular IDEs (d = 5 mm), (**b**) triangular IDEs with three fingers (d $\approx$ 11 mm), and (**c**) triangular IDEs with five fingers (d $\approx$ 5 mm) during IRE.

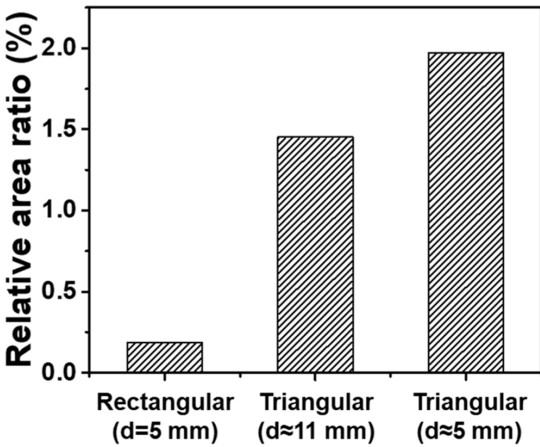

**Figure 6.** Comparison of the relative area ratio of PEI > 0.90 in rectangular IDEs (d = 5 mm), triangular IDEs with three fingers (d $\approx$ 11 mm), and triangular IDEs with five fingers (d $\approx$ 5 mm).

## 4. Discussion

Catheter-based IRE is a promising tool for effective treatment of gastrointestinal tumors, and various electrode configurations, including needle-type, basket-type, and interdigitated electrodes, have been studied. Among them, IDEs have the advantages of allowing for thinner and smaller electrodes and forming a high electric field at a low voltage. Although several studies have been performed to evaluate IRE characteristics according to the design parameters of rectangular IDEs, relatively few studies have been conducted on the electrode configuration of IDEs.

Triangular IDEs allow strong electric fields to be induced at the tip of the electrode fingers, and can deliver electric field energy to the tissue more efficiently than conventional rectangular IDEs. Therefore, triangular IDEs can provide an alternative catheter-based IRE electrode design.

In general, about 70–100 pulses are needed as a threshold for almost-complete tissue ablation for IRE [14,36]. However, due to the limitation of computation capacity, we only delivered a set of 10 pulses with a duration of 100 μs and a frequency of 1 Hz to the tissue. These simulation conditions act as a limitation in examining overall tissue ablation by IRE; however, it is deemed sufficient to compare the electric field distribution according to the finger shape of the IDEs and provides a new alternative in the electrode design of catheter-based IRE.

## 5. Conclusions

We propose and demonstrate triangular IDEs that can deliver electric field energy more efficiently than conventional rectangular IDEs and cause injury to a desired tissue. A strong electric field was induced at the tip of the triangular IDEs. The relative area ratio of the triangular IDEs increased by approximately 10 times compared to that of conventional rectangular IDEs at the same electrode distance. We believe that these results will be very helpful for designing electrodes in catheter-based IRE devices.

**Author Contributions:** Conceptualization, D.-J.L. and D.Y.K.; methodology, D.-J.L.; writing—original draft preparation, D.-J.L.; writing—review and editing, D.-J.L. and D.Y.K. All authors have read and agreed to the published version of the manuscript.

**Funding:** This work was supported by the INHA UNIVERSITY Research Grant.

**Institutional Review Board Statement:** Not applicable.

**Informed Consent Statement:** Not applicable.

**Data Availability Statement:** Not applicable.

**Conflicts of Interest:** The authors declare no conflict of interest.

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
