# Peer review of "Enhanced Electrical Injury Using Triangular Interdigitated Electrodes for Catheter-Based Irreversible Electroporation"

_applsci, doi:10.3390/app13148455_

Round 1
Reviewer 1 Report
“Enhanced electrical injury using triangular interdigitated electrodes for catheter-based irreversible electroporation” by Dong-Jin Lee and Dae Yu Kim describe COMSOL simulation of triangular interdigitated electrodes (IDEs) with the objective to create stronger electrical field than rectangular IDEs.
Contribution is well structured, but the presented results are too little for publication in a journal. There are countless articles analysing and studying interdigital electrodes for a wide variation of application. Moreover, all the equations and parameters are used from reference [14]. So, this contribution needs more scientific content specifically for this irreversible electroporation.
Some comments/ questions:
Introduction
1) “Although the geometric parameters and configuration of the electrodes are closely related to the outcomes of IRE, there is a relatively lack of research on the evaluation of IRE characteristics for the role and impact of electrode configurations.” About what are you referring to ‘lack of research’?
Section 2.3
2) “The conductivity-temperature coefficient was taken as 0.02 (1/?) in this study.” How is this value, 0.02, chosen?
Section 3
3) Table 1: Is ‘A’ value also taken from reference [14]?
4) “The ranges of the electrical conductivity change of the ML, SL, and MP were 0.511–1.638, 0.251–0.803, and 0.202–0.586 S/m, respectively.” I assume these values are also from reference [14]? Are these values also used in COMSOL simulation for electrical field and for probability of electrical injury to see their influence?
5) There is no discussion in the text for the results shown in Figure 2; with other words what information is obtained from Fig.2 that is useful for further research and the following results?
6) “It can be seen that the voltage change occurs rapidly at the edge of the triangular IDEs compared to that of the rectangular IDEs.” It is not easily to see a big difference in voltage profile between a9 and c) in Fig. 3. So, please make clear quantitatively this difference you are mentioning.
7) “For the triangular IDE, a strong electric field was observed at the tip region and the maximum magnitudes were about 2.54×106 V/m at d≈11 mm and 3.17×106 11 V/m at d≈5 mm.” Please make a short explanation why this difference.
8) Electrical field is proportional to the distance between electrodes. This cannot be seen in Fig. 4. for 5cm vs 11cm. Can you please explain?
9) As ML/ SL / MP are thinner than 4 mm, I do not see electrical field penetration through these layers and even below them. Why is this not seen in the simulation? How much of the electrical field is ‘absorbed’ in the tissue?
10) What is ‘Relative area ratio of PEI’? Not defined in the text.
11) “.....(RAR) of PEI > 0.90 was calculated as 19 shown in Fig. 6.” Do you mean ‘shown’ as it is not calculated.
English is ok.
Reviewer 2 Report
Lee et al. described the electrical property of a new IDE form for the potential use of tumour therapy via cellular membrane structural disruption. Overall the work is interesting and simple. There are some small improvements the author should consider to improve the manuscript's readability.
1. The discussion is too short to be a meaningful section. It presents redundant information in the intro and the conclusion section. Please reconsider rewriting or restructuring the paper accordingly.2. Environmental factors such as pH or ionic strength can also change the electrical field affecting downstream IRE results; the author may consider regenerating the modelling on different conditions
English style and language looks good, please perform checks for minor error then it should be suitable for publication.
Round 2
Reviewer 1 Report
Thank you for your good update of the contribution.
It will improved a bit more if you explain the results shown in figure 2, what information is to be taken from this figure.
Minor editing of English language required.